# Back Pain in Rare Diseases: A Comparison of Neck and Back Pain between Spinal Cord Ischemia and Spinal Dural Arteriovenous Fistula

**DOI:** 10.3390/brainsci10090618

**Published:** 2020-09-07

**Authors:** Anca Elena Gogu, Agneta Pusztai, Alina Zorina Stroe, Daniel Docu Axelerad, Any Docu Axelerad

**Affiliations:** 1Department of Neurology, University of Medicine and Pharmacy “Victor Babes”, 300041 Timisoara, Romania; agogu@yahoo.com; 2Department of Anatomy, University of Medicine and Pharmacy “Victor Babes”, 300041 Timisoara, Romania; agipusztai@yahoo.com; 3Department of Neurology, “Ovidius” University, General Medicine Faculty, 900470 Constanta, Romania; docuaxi@yahoo.com; 4Department of Sport, Faculty of Physical Education and Sport, “Ovidius” University, 900470 Constanta, Romania; docuaxy@yahoo.com

**Keywords:** neck and back pain, VAS score, MR- angiography

## Abstract

Neck and back pain may be noted like a first symptom in rare diseases: spinal cord ischemia and spinal dural arteriovenous fistula (SDAVF). Spinal cord ischemia is a rarer pathology, compared with cerebral ischemia, yet the morbidity and mortality are comparable in both cases; furthermore, classifying the acute loss of function in the spine, encountered in spinal cord ischemia as an important neurological entity. SDAVF presents the same clinical symptoms as spinal cord ischemia, but even though it has a progressive character, the impact in the quality of patients’ lives being equally as important. Between August 2012–August 2017 we admitted through the hospital emergency department 21 patients with spinal cord ischemia and 11 patients with SDAVF (only self-casuistry). Demographic (age, gender), clinical, imagistic (Magnetic Resonance Angiography, Magnetic Resonance Imaging), paraclinical data as well as history, time to diagnosis, the visual analogue scale for pain (VAS score), risk factors, surgical and medical treatment, evolution, neurorehabilitation, were all used to compare the two lots of patients. The aim of this study was to observe potential differences in the demographics, symptomatology, VAS scores and treatment in comparison for spinal cord ischemia and SDAVF, to facilitate the further recognition and management in these diseases. In group A we have 21 patients with spinal cord ischemia (14 females, 7 males). The median age was 41.3 years (range 19–64). The median time to diagnosis was 7 h. The most frequent symptoms were acute neck or back pain at onset (100%), motor deficits (95.24%), sensory loss (85.72%), and sphincters problems (90.48%). The most common location was the lumbosacral spine (14 cases; 66.67%; *p*-value = 0.03) for spinal cord ischemia and the thoracic spine (7 cases, 63.64%; *p*-value = 0.065) for SDAVF. The treatment of spinal cord ischemia was medical. In group B we included 11 patients (6 females, 5 males). The median age was 52.6 years (range 28–74). The median time to diagnosis was 3 months (range 2 days–14 months). Patients have progressive symptoms: neck or back pain (100%), gait disturbances (100%) and abnormalities of micturition (100%). The treatment of SDAVF was surgical occlusion of fistula. The proportion of severe VAS score (7–10) in patients with spinal cord ischemia was significantly higher than that in patients with SDAVF (100% vs. 18, 19%; *p*-value = 0.051). Taking into consideration that the usual findings and diagnosis of spinal cord ischemia and SDAVF are still challenging for neurologists and in some cases the difficulties are related to technical limitations, we consider these entities to be rare but very important for the life of our patients. Patients were grouped into spinal cord ischemia and SDAVF status and those with acute or chronic pain conditions, measured by the VAS score. Patients with spinal cord ischemia develop acute neurological symptoms. They are much younger than the patients with SDAVF and the recovery rate is higher. Patients with SDAVF develop a progressive myelopathy and they suffer considerable neurological deficits. Imaging the lesions with MR angiography or MRI, we can confirm the diagnosis.

## 1. Introduction

Spinal cord ischemia and spinal dural arteriovenous fistula (SDAVF) are rare and underdiagnosed diseases. Recent advances in diagnostic imaging procedures allow a greater accessibility to study the spinal cord and its vascular system during lifetime [1].

There is always a correlation between the vascular anatomy of the spinal cord and the clinical symptoms. We know that the spinal cord grey matter is located internally, forming an H-shape, whereas the white matter tracts are located outside. The anterior two-thirds are related to the motor and spinothalamic tracts, while the posterior third contains proprioceptive pathways [1].

Infarction in the distribution of the anterior spinal artery is most often caused by a disease of the parent artery (e.g., the aorta) and less often by an embolism or an intrinsic disease of this artery (atherosclerosis, Lyme borreliosis, neurosarcoidosis, systemic lupus erythematosus).

Acute neck or back pain, weakness, flaccid paresis accompanied by diminished superficial and tendon reflexes below the level of the lesion, sensory loss and sphincter problems were the most common symptoms of the spinal cord ischemia at the time of presentation [2]. Maximum disability is observed within 12 h of onset in a majority of patients [3]. The medical treatment of spinal cord ischemia is generally supportive, and includes maintenance of blood pressure, reversal causes such as hypovolemia and arrhythmias, antithrombotic therapy, antalgic therapy, early bed rest.

Spinal vascular malformations should be divided into two large groups, which have different blood supplies, presentations and clinical findings: the dural (type 1) and intradural (type 2) groups [4,5,6]. SDAVF (type 1) is the most common vascular malformation of the spinal cord; it is an abnormal shunt between a spinal radicular artery and corresponding radicular vein that drains the perimedullary venous system [7].

Because it presents the same clinical symptoms as spinal cord ischemia, SDAVF is another entity which must be considered. However, in these patients’ case, the most frequent symptoms are progressive, including pain. Pain can be radicular, sometimes mimicking sciatic pain, often worsening after exercises. Surgical treatment is indicated for SDAVF. Embolization of SDAVF is not possible in our clinic.

Functional recovery after physical and occupational therapy is higher in patients with spinal cord ischemia compared to SDAVF. Chronic pain was a disabling consequence of spinal cord ischemia and SDAVF.

## 2. Materials and Methods

This paper is an on-going study on spinal cord ischemia and SDAVF, the first report was prepared in 2014. Between August 2012–August 2017 we admitted through the hospital emergency department 21 patients with spinal cord ischemia and 11 with SDAVF (only self-casuistry).

Spinal cord MRI with angiographic sequence (MRA) was performed in all patients using a Siemens Medical Systems 1.5-T MR unit. The examined casuistry includes within the MRI (MRA) patterns from the literature. Magnetic resonance imaging (MRI) is the imaging procedure of choice for detecting spinal cord ischemia. The pattern of signal changes and their time course are similar to those for cerebral infarction.

The patients with spinal cord ischemia were diagnosed using MRI. After spinal cord infarction, the typical spin-echo magnetic resonance (MRI) findings are cord enlargement and hyperintense signal on T2-weighted initially, with or without gadolinium enhancement may demonstrate a double-dot (“owl s eye”) pattern in the region of the anterior horns and H-shape pattern involving the central grey matter or a more diffuse pattern involving both grey and white matter [3]. MRI, with contrast-enhanced, three-dimensional magnetic resonance angiography (MRA) is the diagnostic procedure of choice in the initial evaluation of suspected spinal dural arteriovenous fistula.

The patients with SDAVF (type 1) were diagnosed by diffusion-weighted imaging (DWI), T2-weighted images and MRA. DWI often shows spinal cord infarction, increased T2 signal reveals spinal cord edema. The key finding that suggests the possibility of a fistula is serpiginous dilated veins on a surface of the spinal cord (“flow-void phenomena”). These are now often seen on T2-weighted and gadolinium-enhanced images and on contrast-enhanced MRA. We have information on history, time to diagnosis, neurological examination, neck and back pain scale (the The Visual Analogue Scale for Pain score), MRI and MRA, treatment and neurorehabilitation. The Visual Analogue Scale for Pain (VAS score) is the optimal tool for describing pain severity or intensity. This scale pain scale represents a simple assessment tool consisting of a 10 cm line with “0” for no pain and “10” for the worst pain ever experienced. The researchers found that absolute values of pain on a “0–10” scale naturally grouped into three categories: 1–4 mild pain; 5 or 6 moderate pain and 7–10 severe pain. We determined the VAS score at onset, pre-and after medical or surgical treatment and after three months of evolution.

We re-examined the patients after three months and compared their current status: neck or back pain, walking, paresthesias, sphincter problems, muscle spasm as worse, same or better.

### Statistical Analysis

Medical statistics is a branch of statistics which focuses on medical applications. As an input, a collection of statistical data from 2017 has been used, containing software packages of medicine and health sciences, including epidemiology, public health, forensic medicine, and clinical research. Dataset used was hypothesis driven research.

For the computation of the test statistic and the critical value, ANOVA (analysis of variance; single factor; two-factor with replication) tool was used as well as QI Macros Excel (a statistical process control software package plugin for Microsoft Excel) to make Hypothesis Testing. Post-hoc tests (Tukey’s Honestly Significant Differences (HSD)) were conducted together with ANOVA to determine which groups differ from one another. With this specific post-hoc test all possible pairwise comparisons were done.

The null hypothesis was a hypothesis of no difference between demographics, clinical features and scores in the tests performed between group A and group B.

The alternative hypothesis is the opposite of the null hypothesis and was the hypothesis set out to investigate. If *p*-value is less than the chosen significance level, then the null hypothesis is rejected.

For a better interpretation of the resulting values, in ANOVA it was considered that if *p*-value < 0.05 the values described are statistically significant. It indicates strong evidence against the null hypothesis, as there is less than a 5% probability the null is correct. If *p*-value > 0.05 the values are not statistically significant and indicates strong evidence for the null hypothesis. This means we retain the null hypothesis and reject the alternative hypothesis.

## 3. Results

Patients were grouped into spinal cord ischemia and SDAVF status. In group A we have 21 patients with spinal cord ischemia (14 females, 7 males; 66.67% vs. 33.33%). The median age was 41.3 years (range 19–64). The median time to diagnosis was 7 h.

Disease of the aorta is the most commonly recognized cause of spinal cord ischemia. We have two patients, males, with abdominal aortic aneurysms. After direct open aortic aneurysm surgery, they presented neurological complications. One of them, with thoracic spinal cord ischemia presented paraplegia, urinary incontinence and spinothalamic type sensory deficits. The other one, with lumbar spinal cord ischemia presented unequal paraparesis, loss of sphincter function, areflexia of the legs, loss of touch sensation in the lower limbs. At the onset of spinal cord ischemia, the VAS score was 10 for both patients and they presented acute back pain.

Embolism can cause spinal cord ischemia. We have a patient, male, with atrial myxoma who developed thoracic spinal cord ischemia. The VAS score was 9.

Another cause of spinal cord ischemia was cervical intervertebral disk degeneration in three young women with thrombophilia and on oral contraceptives treatment. All of them have the primary hypercoagulable status: factor V Leiden mutation, prothrombin G 20210A mutation, MTHFR gene mutation. The first symptom was strong pain in the neck or upper back (VAS score = 10), with radicular distribution, before neurological signs: asymmetric quadriparesis, spinothalamic tract sensory loss and urinary retention. The immediate precipitant factors were: neck massage, repetitive minor trauma and brutal neck motion.

We described the most important cases.

### 3.1. Case 1A

Syringomyelia-like with central spinal cord infarction at cervical levels C4–C7, in 28-year old women with acute onset: strong pain in the upper limbs and in the neck (VAS score = 10), dissociated sensory loss, with loss of pain and temperature under the level of the lesion, but preserved touch sensation, with spared posterior column and motor function. Imaging data is available in Figure 1. 

Spinal cord ischemia may also develop during acute and chronic herniated vertebral disks with compression at the lumbar spinal cord or nerve roots. We have nine cases (6 females, 3 males) with acute onset of severe spinal cord dysfunction.

### 3.2. Case 2A

A 42-year old male, with acute back pain (VAS score = 9) and in the lower limbs, areflexia of the legs, loss of sphincter function and spinothalamic type sensory deficits under the level of the herniated vertebral disk (L2). Imaging data is available in Figure 2a,b.

We described two cases of Lyme borreliosis disease, females with thoracic/lumbar spinal cord ischemia caused by infectious spinal arteritis. The cerebrospinal fluid was abnormal and specific antibodies to Borrelia were present in the blood and cerebrospinal fluid.

### 3.3. Case 3A

A 49-year old female, with acute back pain (VAS score = 8), paraplegia, sensory loss under T8 level, micturition disturbances. She was diagnosed with Lyme disease about one year ago. Borrelia burgdorferi antibodies were present in her blood at onset of the spinal cord ischemia (Immunoglobulin G = 21.60 UA/mL, negative UA/mL < 10; Immunoglobulin M = 11.30 UA/mL, negative UA/mL < 18). Imaging data is available in Figure 3a,b. 

Sarcoidosis causes a variety of central nervous system manifestations, including intraparenchymatous granulomas, meningitis and myelopathy [1]. We have two cases, one male and one female, with angeitic neurosarcoidosis, a disorder that can also affect the spinal cord and peripheral nervous system. They have spinal cord infarction at the lumbar level and conus medullaris, with back pain (VAS score = 9), paraparesis and micturition problems. MRI shows expansion of the lumbar spinal cord with intramedullary regions of nodular enhancement after gadolinium. The symptomatology responded to corticosteroids in substantial doses and over long periods (e.g., 70 mg of Prednisone daily for 6 months). We have used immunosuppressant drugs such as Azathioprine for one year. We have another case, a young female, with acute onset: asymmetrical paraparesis, sensory symptoms, back pain (VAS score = 9) and micturition disturbances. She had a spinal cord ischemia at the thoracic levels T6-T12 and she was known with systemic lupus erythematosus (SLE) for about six years. Nervous system findings are common in patients with SLE. The infarcts are of diverse causes: small vessel disease, abnormalities of coagulation and cardiac origin embolism.

The most frequent symptoms were: acute neck or back pain at onset (100%) with the VAS score between 9 and 10 (severe pain), gait disturbances (tetraparesis/tetraplegia or paraparesis/paraplegia) (95.24%), sensory loss (85.72%), sphincter problems (90.48%). The most common location to be affected is the lumbosacral cord (14 cases, 66.67%). The lower cervical spinal cord was involved in 4 cases (19.05%). We have 3 patients with midthoracic spinal infarction (14.29%). Medical treatment modalities: anti-inflammatory, antalgic therapy (Pregabalin, Gabapentin, Duloxetine), for three months, etiologic treatment.

### 3.4. Group B

In group B we have 11 patients (6 females and 5 males; 54.55% vs. 45.45%) with SDAVF. The median age was 52.6 years (range 28–74). The median time to diagnosis was 3 months (range 2 days–14 months). The most frequent presentation of SDAVF is that of progressive neurological symptoms with acute deteriorations. Pain was present in all patients, sometimes like sciatic or radicular pain, aggravated by exercise and physical activity. The VAS score was between 1–4 (mild pain) in 3 patients (27.28%) and between 5 and 6 (moderate pain) in 6 patients (54.55%). Only 2 patients had severe pain (VAS score 9) comparative with patients with spinal cord ischemia (18.19% vs. 100%; *p*-value = 0.051). The most frequent symptoms involve gait (paraparesis/paraplegia-100%), sensory abnormalities in the lower limbs or/and perineum (100%) and problems with micturition, defecation and sexual function (100%).

Most fistulas are solitary and involve the lower thoracic and lumbosacral segments. We have 7 patients (63.63%) with thoracic fistula: 2 cases at T3-T6, 4 cases at T7-T9 and 4 patients (36.37%) with lumbosacral fistula: 2 cases at L1-L2 and 2 cases at L4-L5. There is statistically significant difference between lumbosacral localization in patients with spinal cord ischemia versus SDAVF (66.67% vs. 36.37%; *p*-value = 0.03). Only one patient had type 1 spinal vascular malformation at level T2-T3 with epidural hemorrhage (Case 1B).

### 3.5. Case 1B

Patient was a 62-year old male, with progressive onset in four months, who presented initial symptoms such as difficulty in climbing stairs, paresthesias and radicular pain in the lower limbs. The symptoms were ascending in time and later patient was paraplegic, with total sensory loss below T2, urinary retention and constipation. Imaging data is available in Figure 4a,b.

### 3.6. Case 2 B

A 32-year old female, with subacute onset in three weeks, presenting with asymmetrical paraparesis, spinothalamic type sensory deficits. Posterior column sensory functions (vibration and position sense) are spared. Micturition disturbance was present. Back pain was present at onset with VAS score 6 (moderate pain). Imaging data is available in Figure 5. 

## 4. Treatment Modalities

Treatment modalities: the aim of treatment in SDAVF is to occlude the shunting zone (the most distal part of the artery together with the most proximal part of the draining vein) [8,9,10].

There are two options for the treatment of SDAVF: surgical occlusion of the intradural vein that received the blood from the shunt zone or embolization of fistula. In our clinics, this last procedure is not possible.

Patients’ variables from demographic data, symptomatology, localization of the lesion and the VAS score were summarized and compared in Table 1**.**

We re-examined the patients after three months and compared their current status: neck or back pain (VAS score), walking disturbances, muscle power and muscle spasms, paresthesias and sphincter problems (Table 2).

Recovery is higher in patients with spinal cord ischemia. Nineteen patients (90.48%) had a favorable outcome: they could walk with one assistive device or none and no need for urinary catheterization after the subacute phase of the disease. After three months, all of them walked without assistive devices.

The effect of surgical treatment in patients with SDAVF on activities of daily life was reported as better by 8 patients (72.73%) and worse by one. Three patients were severely affected and needed a wheelchair. The most persistent symptoms were paresthesias, muscle spasms, micturition problems and pain.

Neurorehabilitation is an important goal for many patients seeking physical therapy intervention. Enabling individuals to manage daily self-care is among the most important goals undertaken by the rehabilitation team. Self-care tasks include eating, dressing, bathing, grooming, use of the toilet, mobility within the home and represent the general category of activities of daily living (ADL).

A few basic exercises were used in this study: passive/passive-active movements in all diagrams, proprioceptive neuromuscular facilitation, rhythmic stabilization, contract-relax, hold-relax, repeated contractions, bridging, quadruped posture, sitting, sit-to stand, modified plantigrade, standing, walking (with ambulatory assistive devices), ascending stairs and descending stairs.

## 5. Discussion

Stroke located in the spinal region represents a small fraction of all patients with central nervous system vascular disease but continues to present a challenge for diagnosis and treatment. Clinicians recognized that most spinal cord infarcts involved the anterior portion of the spinal cord by anterior spinal artery occlusion.

Spinal cord ischemia is most often caused by the disease of the parent artery (aortic aneurysm) and less often by embolism (atrial myxoma, bacterial and non-bacterial thrombotic endocarditis). Intrinsic disease of intraspinal arteries is much less common (Lyme borreliosis, central nervous system vasculitis like neurosarcoidosis, systemic lupus erythematosus). Since the early 1980s, pathologists and clinicians have become aware that cartilaginous material from intervertebral disks can somehow invade the spinal arteries and veins and cause devastating spinal cord strokes [11,12,13,14,15]. Most reported cases are cervical and involve young women on oral contraceptives [11,15]. Between the subjects of the study, are encountered three young women with cervical intervertebral degeneration, thrombophilia and oral contraceptive treatment. Also, our study has included nine patients who developed spinal cord ischemia during acute or chronic herniated vertebral disks with compression at the lumbar spinal cord or nerve roots [16,17,18,19,20,21,22,23,24,25]. Most spinal cord ischemia are located in the lumbosacral region (14 cases; 66.66%). In patients with SDAVF, only four cases (36.37%; *p*-value = 0.03) had lumbosacral lesions. When the lumbar cord is involved, a conus medullaris infarct develops with motor deficits, in the lower limbs, wasting and areflexia of the legs, loss of sphincter function, variable sensory loss, back pain or pain in legs. The VAS score of these patients was between 7–10 (severe pain).

Because it presents the same clinical symptoms as spinal cord ischemia, SDAVF is another entity that must be considered. Patients, who are mostly middle-aged (median age 52.6 years), develop a progressive myelopathy which at the early stage of the disease often mimics polyradiculopathy or anterior horn cell disorder. By the time involvement of upper motoneurons or sacral segments makes the diagnosis of SDAVF inescapable, patients suffer from considerable deficits [26,27,28,29,30]. The most frequent symptoms were neck and back pain, gait disturbances, sensory loss and urinary problems. Most SDAVFs are located in the thoracic region (7 cases; 63.63%). In patients with spinal cord ischemia, only 3 cases (14.29%; *p*-value = 0.065) had thoracic lesions. The back pain in patients with SDAVF was moderate (VAS score between 5–6). The essential investigations to establish the diagnosis of spinal cord ischemia and SDAVF are MRI and MRA, which should be performed when an acute or progressive myelopathy is suspected. MRI findings include hypo-intensities on T1-weighted images and hyper intensities on T2-weighted images [30,31,32,33,34,35]. Increased signal intensity in the center of spinal cord and peripheral sparing on T2-weighted images is found in all patients. In all patients with SDAVF, MRA reveals flow in serpentine perimedullary structures and indicated the level of the fistula. Only one patient had SDAVF at level T2-T3 with epidural hemorrhage.

Recovery is increased in patients with spinal cord ischemia. This recovery is an intrinsic person/nervous system process, dependent on psychological, behavioral, neurophysiological and time-related factors. The role of neuromuscular rehabilitation is to optimize the recovery of movement control, working with all these factors [2]. There are three main principles to consider in neuromuscular rehabilitation: functional movement, skill-and ability level rehabilitation and the code for neuromuscular adaptation.

A limitation of the present study could be related to the small number of the research participants. The low prevalence of spinal cord ischemia and SDAVF comparative with stroke located in the brain is another limitation of this study but continues to present a challenge for diagnosis and treatment. This study analyzed only patients with neurological symptoms. The VAS score is a subjective criterion and could also be a limitation. Neurorehabilitation was carried out in different locations. Futures studies, in larger cohorts, includes more clinical and imagistic variables, should these results be validated.

## 6. Conclusions

Spinal cord ischemia and SDAVF are poorly diagnosed because of both, technical limitations and lack of neurological experience in the subject, we consider these entities to be rare but very important for the life of our patients.

In our study, the patients’ ages between the onsets of the diseases was statistically significant, the patients with spinal cord ischemia being younger than the patients with SDAVF (median age was 41.3 years vs. 52.6 years; *p*-value = 0.0371).

In spinal cord ischemia was observed a prevalence of female gender that were affected compared with males (14 cases; 66.67% vs. 7 cases; 33.33%).

The median time to diagnosis of spinal cord ischemia was only a few hours (under 7 h) but for patients with SDAVF, this time is longer (weeks or months).

The diagnosis of spinal cord ischemia was confirmed by MRI which reveals spinal cord enlargement and hyperintense signal on T2-weighted images in all patients. Gadolinium enhancement demonstrated a double-dot (“owl’s eye”) pattern in the region of the anterior horns. The most common location to be affected by ischemia is the lumbosacral spine.

Patients with spinal cord ischemia develop acute neurological symptoms. All of them had a VAS score at onset between 7–10 (severe pain). The onset of the neurological symptoms (motor deficits, sensory loss, sphincter problems) occurred in a few minutes. Meanwhile, the patients with SDAVF develop progressive myelopathy and they suffer considerable neurological deficits. In our experience, most SDAVFs are located in the thoracic region (T2-T9). Most of the patients with SDAVF had a VAS score at onset between 1–4 (mild pain) and 5–6 (moderate pain). Another interesting fact discovered in our study is related to the VAS score was not statistically significantly different between the two groups.

The diagnosis of SDAVF was confirmed by MRA which reveals “flow-void phenomena”, representing tortuous and dilated vein at the dorsal surface of the spinal cord. The time from the debut of the symptoms to diagnosis was between two days and 14 months with an average of three months. The diagnosis of SDAVF was established in a few hours to one day after hospitalization, and the neurosurgical intervention was performed in an average of 2–3 days after diagnosis because of the technical reasons. The establishment of the neurosurgical treatment from diagnosis was not performed late in our opinion; but certainly, in some cases, the interval from the occurrence of the symptoms to the neurosurgical treatment was prolonged, bringing major inconveniences in the patient’s evolution. The majority of the patients in the SDAVF group presented cord infarction and ongoing venous congestion. Endovascular examination was difficult due to technical imputations.

Surgical procedures were technically successful in patients with SDAVF. Embolization is not possible to be performed in our clinics. Postoperative control of fistula in patients from the SDAVF group was performed by postoperative MRA as a routine procedure in the neurosurgery clinic, fact confirmed by the clinical evolution of patients. In 1–2 cases from the SDAVF group, the postoperative control of fistula was performed by digital subtraction angiogram. The MRA postoperative control was chosen because it is a non-invasive procedure compared to digital subtraction angiogram which is an invasive procedure and requires more risks to be taken into consideration.

In an interval of a month after the neurosurgery intervention, patients from the SDAVF group were advised to perform a fistula control imaging procedure, represented by an MRA or a spinal CT angiography, the latter being preferred by our team of neurosurgeons because it presents more conclusive if the fistula has thrombosed spontaneously or has closed completely, but with a with the negative aspect of a risk of large radiation.

Spinal cord ischemia benefits of medical treatment (maintenance of blood pressure, reversal causes, antithrombotic therapy, and early bed rest).

Recovery is higher in patients with spinal cord ischemia, compared with the patients with SDAVF. Our patients with spinal cord ischemia had a favorable outcome, after three months all of them could walk without assistive device and without needs for urinary catheterization. A significant improvement was observed in patients with SDAVF in walking and muscle power. Paresthesias, muscle spasms and sphincter problems persisted after surgical and physical therapy. Chronic pain was a disabling consequence of spinal cord ischemia and SDAVF. Cognition about injury and pain, persistent pain and fear of it, behavioral factors can all be managed within skill-level rehabilitation.

## Figures and Tables

**Figure 1 brainsci-10-00618-f001:**
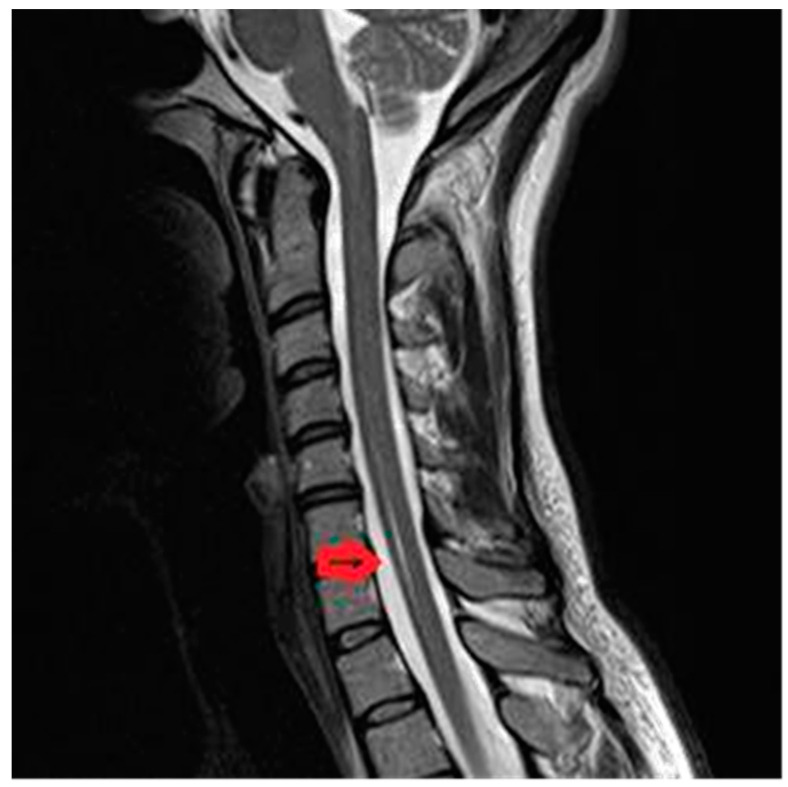
Description: Magnetic Resonance Imaging (MRI) shows a hyperintense signal on T2-weighted Turbo Spin Echo (TSE) sequence at C4–C7 levels.

**Figure 2 brainsci-10-00618-f002:**
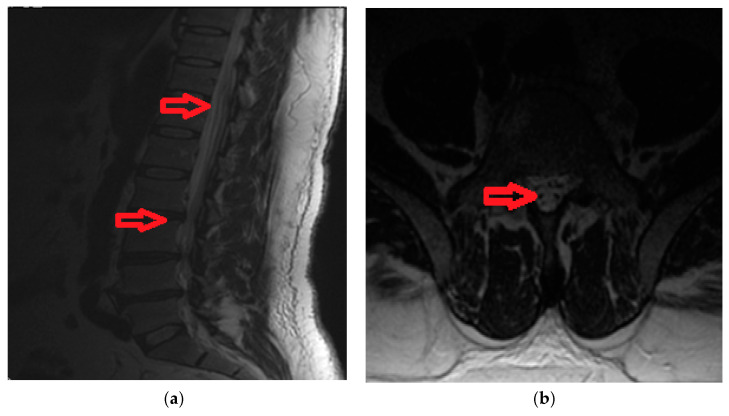
(**a**). Magnetic Resonance Imaging findings include hyperintensities on T2-SAG-FRFSE (T2-weighted Sagital Fast Recovery Fast Spin Echo) images with gadolinium enhancement at T10-L2 levels and herniated disk L2. (**b**) Magnetic Resonance Imaging reveals a double-dot (“owl’s eyes”) pattern in the region of the anterior horns of spinal cord on T2-Ax-FRFSE (T2-weighted Axial Fast Recovery Fast Spin Echo).

**Figure 3 brainsci-10-00618-f003:**
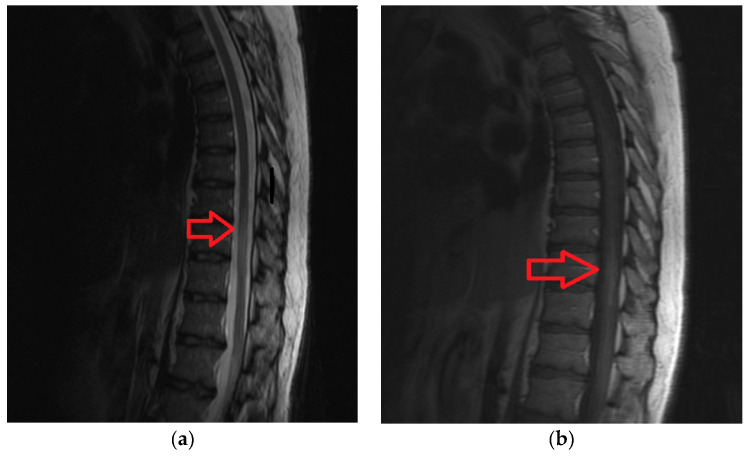
(**a**) Magnetic Resonance Imaging shows a hyperintense signal on T2-SAG-FRFSE (T2-weighted Sagital Fast Recovery Fast Spin Echo) sequence at T8-T10 levels. (**b**) Magnetic Resonance Imaging shows a hypointense signal on T1 -SAG-FSE (T1-weighted Sagital Fast Spin Echo) sequence at T8-T10 levels.

**Figure 4 brainsci-10-00618-f004:**
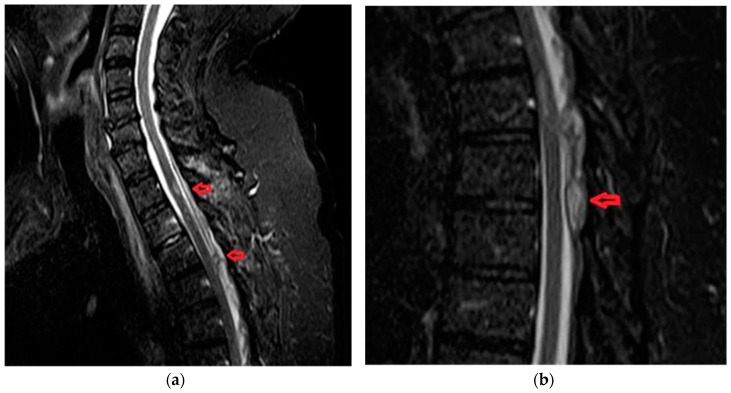
(**a**) MRA shows a “flow voids phenomena” at the dorsal side of the spinal cord below level T2 and multiple zone (T2, T3, T6) with spinal cord edema on T2-weighted sequence. (**b**) MRA revealed posterior extradural acute hematoma at T2-T8 spinal levels.

**Figure 5 brainsci-10-00618-f005:**
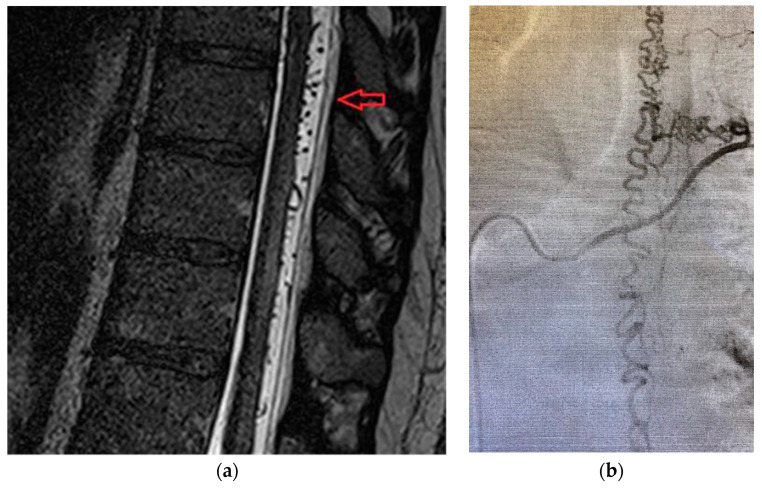
(**a**) Fast-spin-echo-T2-weighted image shows hyperintense cord from T6 to T10 and serpentine flow voids, consistent with enlarged intradural vessels, posterior to the cord from T6 to T10. (**b**) Digital subtraction angiogram (anteroposterior view) following injection of the right T6 posterior intercostal artery demonstrates a fistula in the region of the right neural foramen with drainage into the canal via the medullary vein.

**Table 1 brainsci-10-00618-t001:** Comparison of variables between patients with spinal cord ischemia and SDAVF.

Variable	Spinal Cord Ischemia (*n*,%)	SDAVF (*n*,%)	*p*-Value
Number	21	11	-
Age (years)	41.3 (range 19–64)	52.6 (range 28–74)	0.0371
Gender:	-	-	-
Male	7 (33.33%)	5 (45.45%)	0.46
Female	14 (66.67%)	6 (54.55%)	0.28
Time to diagnosis	7 h	3 h	-
Neck and back pain	21 (100%)	11 (100%)	-
Motor deficits	20 (95.24%)	11 (100%)	0.035
Sensory loss	18 (85.72%)	11 (100%)	0.024
Sphincter problems	19 (90.48%)	11 (100%)	0.048
Localisation of lesion:	-	-	-
Cervical	4 (19.05%)	0	0.24
Thoracal	3 (14.29%)	7 (63.63%)	0.065
Lumbosacral	14 (66.66%)	4 (36.37%)	0.03
VAS score	-	-	-
Mild pain (1–4)	0	3 (27.28%)	0.57
Moderate pain (5–6)	0	6 (54.53%)	0.45
Severe pain (7–10)	21 (100%)	2 (18.19%)	0.051

Abbreviations: SDAVF, spinal dural arteriovenous fistula; VAS, visual analogue scale for pain.

**Table 2 brainsci-10-00618-t002:** The symptoms after three months of onset at patients with spinal cord ischemia and SDAVF.

Rating	Neck and Back Pain	Walking Disturbances	Muscle Power	Muscle Spasms	Paresthesias	Sphincter Problems
Group A spinal cord ischemia = 21						
Worse	0	0	0	0	0	0
Same	3	2	2	2	2	0
Better	18	19	19	19	19	21
Group B SDAVF = 11						
Worse	1	1	2	2	3	1
Same	2	2	2	3	3	1
Better	8	8	7	6	5	9

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
