# Peer review of "Back Pain in Rare Diseases: A Comparison of Neck and Back Pain between Spinal Cord Ischemia and Spinal Dural Arteriovenous Fistula"

_brainsci, 2020, doi:10.3390/brainsci10090618_

Round 1
Reviewer 1 Report
“Back Pain in Rare Diseases: A Comparison of Neck and Back Pain between Spinal Cord Ischemia and Spinal Dural Arteriovenous Fistula”
The manuscript explores the potential of MRI imaging and MRA in the diagnosis of spinal cord ischemia and spinal dural arteriovenous fistula. They compared both clinical conditions by analyzing their pain scores.
Specific comments on weaknesses:
Major points:
- ‘SCI’ is commonly used for ‘spinal cord injury’, using this abbreviation for ‘spinal cord ischemia’ make the whole manuscript very confusing and easy to follow. I suggest using spinal cord ischemia instead of SCI throughout in the manuscript for better understanding.
- Page 1; line 19-21 and page 2, line 67: “SCI causes acute loss of cord function. Because it presents the same clinical symptoms as SCI, but progressive, SDAVF is another entity that must be considered”. I think authors are comparing spinal cord ischemia with spinal cord injury here but used the same abbreviation for both. It is not easy to follow. They need to rewrite the abstract to make it more meaningful for readers.
- The aims and objectives of the study not clearly stated.
- Hypothesis not clearly presented.
- Statistical analysis and type of statistical used for the analysis of data not very clear.
- MRI scanning method and parameters not clearly presented.
- I could not get what authors are concluding from this study, conclusions not clearly presented.
Minor points:
- Page 9 line 262; Nervous system vascular disease but continues to present a challenge for diagnosis and treatment1. It should be treatments (1). There are other places that need to be formatted similarly. Like page 9, line 298: all these factors2. There are three main principles to consider in neuromuscular rehabilitation:
- “Because many neurologists are inexperienced with the usual findings and diagnosis of SCI and SDAVF, we consider these entities to be rare but very important for the life of our patients”. Authors have used this sentence more than once, spinal cord ischemia and DSAVF is poorly diagnosed because of technical limitations, or is a problem of physicians not well experienced?
- Figure legends needs are not very clear
- In figure 2B, both ‘a’ and ‘b’ images are not clear and legend not clear
Author Response
Thank you very much for your help, suggestions and review report!
Please see the attachment.

Reviewer 2 Report
Thank you for your report on SCI versus SDAVF. I am surprised to read patients with SCI have a better clinical outcome that patients with SDAF b/c in my experience patients treated for SDAVF in a timely matter have their venous congestion halted and subsequent improvement in cord signal change. What was the average time to treatment in your SDAVF group... were they treated late and potentially had cord infarction + ongoing venous congestion? Did the treated group get post op images to show that the fistula was fully treated? How were the post operative patients imaged... catheter angiogram?
Thank you for considering these two points.
Author Response
Dear reviewer,
Thank you very much for your support and review report!
„Thank you for your report on SCI versus SDAVF. I am surprised to read patients with SCI have a better clinical outcome that patients with SDAF b/c in my experience patients treated for SDAVF in a timely matter have their venous congestion halted and subsequent improvement in cord signal change. What was the average time to treatment in your SDAVF group... were they treated late and potentially had cord infarction + ongoing venous congestion? Did the treated group get post op images to show that the fistula was fully treated? How were the post operative patients imaged... catheter angiogram?
Thank you for considering these two points. „
- Page 12; lines 359-362: „The patients presented to the neurologist relatively late and the endovascular examination could not be performed in a timely manner. For this reason, patients had spinal cord infarction and adjacent venous congestion, which led to a worsening of the prognosis.”
- Page 12; lines: 366-368: „Postoperatory, a control angiogram was performed to the treated group by the team of neurosurgeons, which showed the closure of the fistula, which was also confirmed by the evolution of the patients. „
Round 2
Reviewer 1 Report
The manuscript has been significantly improved and authors has clearly addressed all the quarries raised by the reviewer.
Author Response
Dear Reviewer,
Thank you very much for your help and your support!
Regards,
The Authors.
Reviewer 2 Report
Thank you to the authors for their revised manuscript.